# Barriers and facilitators of provision of telemedicine in Nigeria: A systematic review

Osagie Kenneth Cole[1,2], Mustapha Muhammed Abubakar[3,4], Abdulmuminu Isah[4,5], Sule Hayatu Sule[6], Blessing Onyinye Ukoha-Kalu[4,7]*

1 Directorate of Clinical Services, Medical Services Branch, Nigerian Air Force, Abuja, Nigeria, 2 School of Public Health, Faculty of Medicine, University of Port Harcourt, Rivers State, Nigeria, 3 Directorate of Therapeutic Services, Medical Services Branch, Nigerian Air Force, Abuja, Nigeria, 4 Person-Centered HIV Research Team, Department of Clinical Pharmacy and Pharmacy Management, University of Nigeria, Nsukka, Enugu State, Nigeria, 5 Department of Clinical Pharmacy and Pharmacy Management, University of Nigeria, Nsukka, Nigeria, 6 Health Commons Solutions Lab, Sinai Health, Toronto, Canada, 7 School of Medicine, University of Nottingham, Nottingham, United Kingdom

* blessing.ukoha-kalu@nottingham.ac.uk

## Abstract

Healthcare access remains a challenge in developing countries and could be a drawback to the attainment of Objective 3 of the Sustainable Development Goals. Digital interventions such as telemedicine have been identified as an effective tool to improve healthcare access. However, evidence suggests that the impact of telemedicine is not uniform globally due to variances in barriers and facilitators. Thus, we conducted a systematic review to identify the barriers and facilitators of telemedicine in Nigeria. The systematic review was pre-registered on PROSPERO (Identification Number: CRD42024609405). Search was conducted on PubMed, Scopus, and the Cumulative Index of Nursing and Allied Health Literature databases. We included studies that reported on the estimates of barriers and facilitators of telemedicine in Nigeria as well as the factors associated with telemedicine implementation, provision, or operation in Nigeria. The outcome was the reportage of barriers and facilitators of telemedicine in Nigeria. A total of 384 studies were identified from the search. After the application of eligibility criteria and deletion of duplicates, 29 studies were included in the review. The most reported barriers were technical and institutional-related while the most reported facilitators were human-resource-related. Technical barriers frequently reported were power outages, poor internet connectivity, and paucity of health professionals with technical expertise while institutional barriers were lack of regulation and poor organizational policies. Formal telemedicine training and education were the most reported human resource facilitators while the use of low-tech educational networks and internet accessibility were the most reported technical facilitators. Findings from this review suggest that technical barriers are a challenge to adopting telemedicine in Nigeria. Evidence shows that education and training

**Data availability statement:** All relevant data are within the manuscript and its Supporting Information files.

**Funding:** The author(s) received no specific funding for this work.

**Competing interests:** The authors declare they have no competing interests.

are critical in addressing these technical challenges. Thus, this review provides a background for interventions towards the effective implementation of telemedicine in Nigeria.

## Author summary

Telemedicine interventions have the potential to improve healthcare access especially in resource-strained settings through the linkage of isolated communities to health services as well as reducing the burden on existing health systems. However, evidence suggests that the observed impact of telemedicine is not uniform across geographical settings. Thus, understanding the peculiar barriers and facilitators of telemedicine in resource-strained settings such as Nigeria would be critical to its provision. This would assist policymakers and healthcare providers to leverage on evidence-based knowledge to provide and enhance the quality of telemedicine interventions. Here, we conducted a systematic review of peer-reviewed articles that reported on the barriers and facilitators of telemedicine in Nigeria. Eligible studies included in the review were 29 papers. Most of the reviewed studies were conducted in the southern part of the country. We found that the most reported barriers were power outages, poor internet connectivity, paucity of health professionals with technical expertise, lack of regulation and poor organizational policies. Formal telemedicine training and education were the most reported facilitators of telemedicine in Nigeria. Our review summarised the evidence on the barriers and facilitators of the effective implementation of telemedicine in Nigeria.

## Introduction

Improved access to healthcare has stagnated since 2015, casting doubt on the attainment of the target of the United Nations Sustainable Development Goal 3.8, which seeks to achieve universal health coverage as well as access to quality healthcare services [1]. Only a quarter of 138 countries achieved the expansion of service coverage, with more than half of 194 countries experiencing worsening or no change in service coverage. The setback in service coverage has been attributed to issues such as geographical inaccessibility, low demand for services, delayed provision of care, low adherence to clinical protocols, and costs [2]. In recognising the role digital health interventions could play in addressing these challenges, the World Health Organization member states collectively approved the World Health Assembly Resolution on Digital Health in 2018 [2]. The WHO defines a digital health intervention as "a discrete functionality of digital technology that is applied to achieve health objectives and is implemented within applications, including communication channels such as text messages". A key category of digital health interventions that enable the provision of health services at a distance is telemedicine. Thus, telemedicine is commonly described as "the delivery of health care services at a distance" [3].

Telemedicine interventions could be in the form of consultations between a remote client and healthcare provider, remote monitoring of a client or diagnostic data, electronic transmission of medical data, and consultations for case management between healthcare providers [3]. These are often enabled by mobile, video, and electronic platforms. Hence, literature around telemedicine has continued to use telemedicine and a broader term, 'telehealth' as synonyms [4].

Significant benefits of telemedicine have been observed and could provide exciting opportunities, especially in resource-strained settings. For instance, telemedicine interventions could facilitate overcoming geographical barriers by connecting isolated persons with healthcare providers and decreasing waiting time in hospitals [5]. It can also reduce the burden on health systems by serving as an intervention to reduce the occurrence of acute events among chronically ill patients [6]. In terms of clinical outcomes, there are strong indications the positive effects of telemedicine care are comparable to in-person care [7,8]. Telemedicine platforms have also been observed to facilitate the transmission of medical images and diagnostic data for prompt remote analysis and interpretations, especially in settings where access is limited [9]. Thus, telemedicine provides opportunities to enhance the coverage of health services if utilized as a complementary intervention to existing health system functions and not as a replacement.

Despite these successes, it has been observed that the impact of telemedicine is not uniform globally. For instance, in sub-Saharan Africa (SSA), a myriad of challenges has been noted to continue to hinder the successful implementation of telemedicine. Challenges noted are typically lack of technological infrastructure, lack of inadequate human resources, and a deficit in digital training [10]. In Africa's most populous country, Nigeria, evidence suggests that some progress has been made concerning the use of telemedicine. These include the gradual involvement of some private and public institutions in utilizing certain telemedicine interventions as well as the recent use of telemedicine in specialist fields such as neurology, endocrinology, and dental surgery [11–14]. Nevertheless, there exist peculiar challenges limiting the nationwide adoption of telemedicine services in the country. First, there exists perennial issues limiting effective telemedicine implementation in Nigeria such as limited internet connectivity and unreliable electricity supply [15–17]. Additionally, Chitungo et al. (2021) noted Nigeria is one of few countries in the continent without an enactment regulating telemedicine, causing a defragmented approach in the utilization of telemedicine across the country. These variances in challenges with other SSA countries translate to dissimilarities in the extent of use and adoption across the region [18].

Therefore, this systematic review aims to understand the barriers and facilitators associated with the provision of telemedicine services in Nigeria. We also aimed to look at the peculiar barriers and facilitators among end users or patients, healthcare providers, and policymakers. A key objective of this project was to thematically synthesise the recommendations of studies that assessed this issue to support key stakeholders in the promulgation of effective telemedicine policies. Representing the first systematic review to examine this issue in Nigeria or any country in SSA, findings from this review could potentially expand coverage of healthcare in Nigeria and provide a background for future interventions to support the effective implementation and scaling of telemedicine services in the continent.

## Methods

### Search strategy

This systematic review was conducted using the Preferred Reporting Items for Systematic Reviews and Meta-Analyses guidelines. The search was electronically conducted in the following databases: PubMed, Cumulative Index of Nursing and Allied Health Literature (CINAHL), and Scopus. The main keywords used were telemedicine and Nigeria. No period limit was applied. Search terms for the keywords were extracted from published systematic and scoping review papers, updated by OKC and MMA, and operationalised using Boolean operators [19–21]. The complete search strategy is at S1 Appendix. This review was registered with PROSPERO (ID Number: CRD42024609405).

## Eligibility criteria

In conducting this review, we applied some inclusion and exclusion criteria during the screening stage. We included articles that were peer-reviewed and written in English. For quantitative studies, we included articles that reported on the estimates of barriers and facilitators of telemedicine in Nigeria. For qualitative studies, we included studies that reported on factors associated with telemedicine implementation, provision, or operation in Nigeria. Also, this study excluded all reviews (scoping, systematic, narrative, and literature), commentaries, editorials, correspondences, and case studies. These sources were excluded because they lacked data needed for systematic reviews and in the case of commentaries, editorials and case studies, primary data is not provided which is required to conduct systematic reviews. We also excluded preprint articles to ensure the reproducibility of this study. Further, studies that did not report on barriers and facilitators as well as factors associated with telemedicine adoption or implementation were excluded. Additionally, interventional, experimental, and quasi-experimental studies that aimed to evaluate the feasibility of implementing telemedicine practice were excluded. Outcomes from quasi-experimental studies could be influenced or explained by other variables, thus, reliability of findings from research work that utilize this study design could be a limitation. OKC and MMA independently performed the screening of titles and abstracts, deleted duplicates, and extracted data to an Excel sheet. Conflicts were resolved by contacting the third reviewer, IA.

## Data extraction, data synthesis, and quality assessment

The following data was extracted from each of the included studies: name of author, year of publication, study area, study population, geographical region, study design, data collection technique, and type of outcome reported. The outcome of the review was the reportage of barriers and facilitators of telemedicine in Nigeria. From all included studies, factors that facilitated or hindered the implementation, provision, and operation of telemedicine were extracted as outcomes. Measure of associations were reported for quantitative, cross-sectional, and mixed methods studies that provided the relationship between telemedicine uptake or utilization and associated factors. We performed a manual thematic analysis to identify relevant themes of facilitators and barriers to telemedicine [22]. We identified the broad themes in the literature and categorised our outcomes under these themes [10,20,23]. Outcomes were analysed under the following sub-headings: human resource related, technical, institutional, and financial. Further, recommendations provided in the included studies were thematically identified and enumerated.

The quality of all included studies was assessed. For cross-sectional and quantitative studies, quality assessment was conducted using the Joanna Briggs Institute (JBI) Critical Appraisal Checklist for Analytical Cross-Sectional Studies. The JBI Tool considers 8 items: inclusion criteria, description of study subject and setting, measurement of exposure, criteria used for measurement of condition, confounding factors, strategies to resolve confounding, and measurement of outcomes as well as correctness of statistical analysis (Moola et al., 2020). For qualitative studies, quality was assessed using the 10-point JBI Checklist for Qualitative Research (Lockwood et al., 2015). We used both tools to assess the quality of mixed-method studies. Responses to each of the items in the quality assessment of both quantitative and qualitative studies could be, 'Yes', 'No', 'Unclear', or 'Not Applicable'. Studies are considered as low quality when there are three reported entries of 'No' or 'unclear'. Four reviewers (OKC, MMA, SHS, BOU-K) independently assessed the risk of bias, and any dispute was resolved by an independent reviewer (IA).

## Statistical analysis

All analyses were conducted in the statistical environment RStudio v.4.2.1. Data in csv format was imported into R Studio for descriptive analysis. Raw data used in this study will be provided upon request.

## Results

A total of 384 studies were identified from three databases. Following the removal of duplicates and the application of our eligibility criteria, 355 papers were excluded. Twenty-nine studies were included in the systematic review [11,12,14,24–49]. Fig 1 is the flow diagram of literature screening.

## Study characteristics

The systematic review comprised a total of 7796 participants across all 29 studies, with the sample size ranging from 3 [29] to 1177 [25]. The study population comprised mainly patients, healthcare providers, and policymakers. Most of the studies (n = 13) focused on healthcare providers [11,24–27,38–40,42,44,46,49,50]. Healthcare providers assessed in the studies were physicians, medical specialists, pharmacists, nurses, and medical students. Eleven studies assessed the experience of end users and patients in using telemedicine [12,14,30,31,33,34,36,41,45,47,48] Patients identified across the studies were hypertensive, neurology, dental surgery, and diabetic patients as well as persons living with HIV (PLHIV) and pregnant women. Study sample in five of the studies were majorly policymakers [28,29,34,35,37]. Policymakers identified were stakeholders, program managers, and administrators.

Eighteen studies, representing 62% of the included articles, employed cross-sectional study design [11,12,14,24,25,27,31,36–41,43,45–47,50]. Twenty-four percent (n = 7) of the studies were qualitative studies [28–30,34,35,42,44]. Three studies [33,48,49] representing 10 percent of the included studies utilized mixed methods

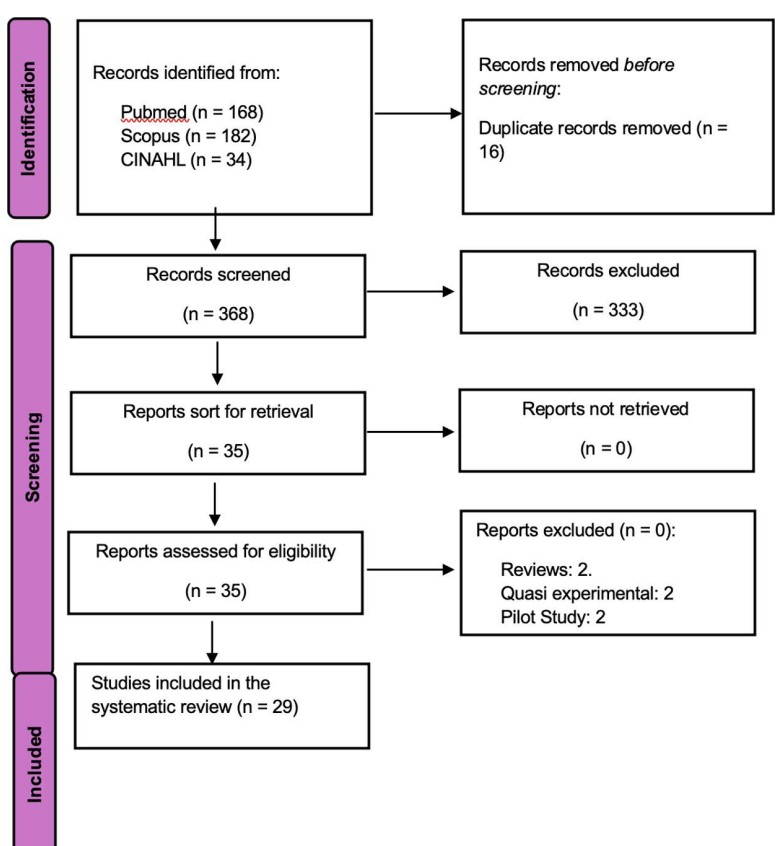

**Fig 1. Flow diagram of literature search using the updated guidelines for reporting systematic reviews by Page et al. (2021).**

study design while one study was quantitative [26]. Only 21 percent of the studies (n = 6) were published before 2020 [24,29,39,40,44,46], with most of the studies published in the last five years.

Fifty-two percent (n = 15) of the studies were conducted in the southern geopolitical regions of the country [14,24,25,29,31,33,36,38,39,42–45,47,50] while 24 percent of the studies were done in the northern part of the country [27,30,34,35,46,48,49]. Twenty-one percent (n = 6) were nationwide or set in different geopolitical regions in both parts of the country [11,26,28,37,40,41]. A detail of the characteristics of all 29 included studies is presented in Table 1.

### Willingness to use telemedicine in Nigeria

Twelve studies reported the willingness to use telemedicine intervention [11,14,31,33,36,39–41,45,47,48,50]. Eight studies reported on willingness to use telemedicine among end users and patients while four studies reported on healthcare providers. Willingness to use telemedicine ranged from 50% [40] to 100% [39]. Among end users and patients, it ranged from 55.7% [48] to 96.2% [31]. Among healthcare providers, it ranged from 50% [39,40] to 100%.

### Barriers and facilitators of telemedicine in Nigeria

The outcome of interest was a report on barriers and facilitators of telemedicine in Nigeria. Eighteen studies reported on both the barriers and facilitators of telemedicine [24–26,30,33–35,38–46,48,49]. Additionally, six studies presented only facilitators of telemedicine [12,14,28,31,36,47] while five studies reported only barriers of telemedicine [11,27,29,37,50]. Table 2 shows the barriers and facilitators reported by all the included studies and the studied populations.

The most reported barriers were technical (n = 8) and institutional (n = 7). Technical barriers were power outages, poor internet connectivity, unfriendly user interphase, lack of software, data security, and lack of technical support. Low mobile ownership was also noted as a technical barrier to telemedicine in Nigeria. Institutional barriers were lack of government support, lack of regulation, absence of school curriculum, and poor organizational policies. Other institutional barriers were administrative rigidity and lack of incentives.

Facilitators were majorly human-resource-related (n = 7) and technical (n = 6). Human-resource-related facilitators were education, formal telemedicine training, experience with telemedicine, and perception of reduced workload associated with the use of telemedicine. Technical facilitators were access to health-related information, the use of low-tech educational networks, and internet accessibility. Other technical facilitators identified were owning a mobile device, ease of use of the platform, and the use of live and recorded interactions. (See Table 3).

**Barriers and facilitators of telemedicine among end-users and patients in Nigeria.** Barriers identified to the use of telemedicine among end-users were financial, human-resource, and technical-related. Patient willingness to pay and cost of service were the reported financial barriers to the utilization of telemedicine service [30,33]. Low literacy level was the only human-resource-related barrier [34]. There were three reported technical barriers to the use of telemedicine among end users namely distrust of technology and privacy concerns, lack of technical support in using service, especially in accessing online status, and low mobile device ownership [34,45,48].

Facilitators of the use of telemedicine among end users and patients were also financial, human-resource, and technical related. Income above 20,000 and 30,000 Naira were the financial facilitators reported in two studies [14,45] Five studies reported human resource facilitators among end users. Human resource facilitators were education [14,36,41,48], previous experience with telemedicine use [36] and patient demography per age and marriage [14,47]. Concerning technical facilitators, five studies reported on factors facilitating the use of telemedicine among patients and end-users. Accessibility to internet connectivity [33,36], owning a mobile device [14], ease of retrieval of health records [30] and using a mix of live and recorded interactions [12].

**Barriers and facilitators of telemedicine among healthcare providers in Nigeria.** Barriers to telemedicine among healthcare providers were institutional, financial, human resource, and technical related. Administrative rigidity [39], lack of incentive to use telemedicine [11], poor organizational and management policies [46], absence in school curriculum

**Table 1. Study characteristics of included papers.**

| First Author | Study Location Reported | Sample Size | Data Collection | Study Design | Sampling technique | Data Analysis & Synthesis |
|---|---|---|---|---|---|---|
| Udenigwe et al [43] | Edo State | 64 | Focus group discussions, interviews | Cross sectional | Purposive | Thematic analysis |
| Olamoyegun et al [14] | Southwest Nigeria | 259 | Questionnaires | Cross sectional | Convenient | Descriptive and logistic regression |
| Kola et al [31] | Ibadan, Oyo State | 260 | Interviews | Cross sectional | NR | Descriptive analysis |
| Akwaowo et al [25] | 4 States in Niger Delta | 1177 | Self-administered questionnaires | Cross sectional | Random | Factor analysis and regression analysis |
| Dele-Ojo et al [47] | Ekiti State | 427 | Questionnaires | Cross sectional | Random | Descriptive analysis |
| Noel et al [27] | Jos, Plateau State | 305 | Self-administered questionnaires | Cross sectional | Stratified | Descriptive statistics, chi-square test, logistic regression |
| Owoeye et al [38] | Benin, Edo State | 340 | Electronic survey | Cross sectional | Multistage | Descriptive analysis |
| Eze & Okojie [29] | Benin, Edo | 3 | Interviews | Qualitative (interviews) | NR | Thematic analysis |
| Archer et al [26] | Nationwide | 12 | Questionnaire | Quantitative | Convenient | Structural equation modelling |
| Olufunlayo et al [37] | Nationwide | 24 | Electronic survey | Cross sectional | Random | Descriptive, non-parametric |
| Ebenso et al [28] | FCT, Ondo, Kano States | 294 | Interviews, document reviews | Qualitative | Purposive | Theory of change, modified TAM |
| Adenuga et al [24] | Ondo States | 252 | Questionnaires | Cross sectional | Stratified | Structural equation modelling |
| Oyetunde et al [40] | Nationwide | 326 | Validated structured questionnaire | Cross sectional | Convenient | Descriptive and multiple linear regression model |
| van Gurp et al [44] | Ibadan, Oyo State | 21 | Unstructured interviews | Qualitative | Purposive | Conceptual mapping |
| Nduka et al [50] | Anambra State | 521 | Questionnaires | Cross sectional | NR | Descriptive and inferential analysis |
| Onigbogi et al [39] | Lagos State | 202 | Pre-tested questionnaires | Cross sectional | Stratified | Descriptive statistics, chi-square test |
| Shekoni et al [42] | Lagos State | 26 | Focus group discussion | Qualitative | Purposive | Braun and Clarke Thematic Analysis |
| Nelissen et al [33] | Lagos State | 236 | Focus group discussion | Mixed methods | NR | Regression analysis |
| Yakubu et al [11] | Nationwide | 48 | Electronic survey | Cross sectional | Convenient | Descriptive analysis |
| Yakubu et al [45] | Ogun State | 398 | Questionnaires | Cross sectional | Random | Descriptive analysis |
| Itanyi et al [30] | Benue State | 35 | Focus group discussion | Qualitative | Purposive | PEN-3 cultural model |
| Obi-Jeff et al [34] | Kebbi State | 144 | Focus group discussion, interviews | Qualitative | Purposive | Deductive approach |
| Iliyasu et al [49] | Kano State | 246 | Structured questionnaires | Mixed methods | Purposive | Logistic regression |
| Obi-Jeff et al [35] | Kebbi State | 135 | Focus group discussion, interviews | Qualitative | Purposive | Thematic analysis |
| Zayyad & Toycan [46] | Northwest Nigeria | 465 | Validated questionnaires | Cross sectional | Cluster, random | Descriptive and correlation coefficient analysis |
| Ojo et al [36] | Ekiti State | 237 | Semi-structured questionnaires | Cross sectional | Systematic | Descriptive and multivariate regression |
| Iliyasu et al [48] | Kano State | 415 | Structured questionnaires, interviews | Mixed methods | Purposive | Descriptive and logistic regression |
| Anibueze et al [12] | Social Media | 294 | Structured questionnaires | Cross sectional | NR | Descriptive statistics, chi-square test, GLM |
| Sarfo et al [41] | Ibadan, Zaria, and Abeokuta | 486 | Structured questionnaires | Cross sectional | Random | Descriptive and logistic regression |

NR: Not Reported; PLHIV: persons living with HIV.

**Table 2. Barriers and facilitators of telemedicine in Nigeria reported in the eligible studies.**

| First Author | Population | Outcome | |
| --- | --- | --- | --- |
| | | **Facilitators** | **Barriers** |
| Udenigwe et al [43] | Ward Development Committee chairpersons and community members | Community involvement, understanding the program, positive attitude, and perceived effectiveness of the program | Limited resources (mobile phone access), shortage of health workers to the program and clash with cultural values of locales |
| Olamoyegun et al [14] | Diabetic patients | Gender (male), married, owning mobile device, education, income above 20,000 Naira | NR |
| Kola et al [31] | Adolescent mothers | Owning mobile device | NR |
| Akwaowo et al [25] | Clinicians | Usefulness, awareness, significant advantage | Data security |
| Dele-Ojo et al [47] | Hypertensive patients | Age (45–64) | NR |
| Noel et al [27] | Medical students | NR | Perception of difficulty to use, poor perception of benefits, low ownership of computers, absence in curriculum |
| Owoeye et al [38] | Health workers | Educated health workers | Lack of ICT facilities, poor internet connectivity, poor electricity supply |
| Eze & Okojie [29] | Administrators | NR | Lack of ICT facilities, poor mobile connectivity, inconsistent electricity supply, lack of political will and govt support |
| Archer et al [26] | Health workers | Privately funded organisations | Technology infrastructure |
| Olufunlayo et al [37] | Health institutions (responses provided by CMDs, CMACs) | NR. | Most institutions had no initiatives in place for domains of processes and regulatory issues across all zones |
| Ebenso et al [28] | 98 end users, 63 health workers and 133 managers/ policymakers | Supportive policy environment, and track record of private-public partnerships facilitated adoption of technology. | NR |
| Adenuga et al [24] | Physicians, nurses | Performance expectancy, effort expectancy, facilitating condition and reinforcement determinants are influential factors in the use of telemedicine services | Willingness to use technology |
| Oyetunde et al [40] | Community pharmacists | Reliability of system | Inability of clients to use system, incessant power outages, system downtime |
| van Gurp et al [44] | Health workers | Low-tech educational networks | Irregular bandwidth, poor network coverage, and unstable power supply obstructing interactivity and access to information |
| Nduka et al [50] | Community pharmacists and patients | NR | Lack of monetary motivation (73.7%), lack of software (56.8%), and operational difficulties (49.3%) were considered major barriers to its implementation |
| Onigbogi et al [39] | Medical doctors | Work duration of five years above, good IT skills, 40–49 years (not statistically significant) | Administrative rigidity |
| Shekoni et al [42] | Health workers | Access to health-related information | Lack of trust, concern of diagnostic accuracy, power and network failure |
| Nelissen et al [33] | Hypertensive patients | Accessibility, attention, and information provision | Patients' unwillingness to pay, understaffing, use of paper records |

*(Continued)*

| First Author | Population | Outcome | |
|---|---|---|---|
| | | **Facilitators** | **Barriers** |
| Yakubu et al [11] | Neurologists | Videoconferencing | Lack of incentive to use tech, poor internet connectivity, lack of exposure to telemedicine |
| Yakubu et al [45] | Neurology patients | NR | Lack of technical support, difficulty assessing status and explaining health conditions |
| Itanyi et al [30] | Pregnant women | Access to free medical screenings, ease of storage and retrieval of health records | Cost |
| Obi-Jeff et al [34] | Program managers (9), community members (124), health workers (10) | For health workers, facilitator was reduced workload while for policy makers, facilitators were involvement of health workers in supporting program, aligning of program with state's priority and reduced workload | Low mobile phone ownership and inability to read messages were barriers to implementation. |
| Iliyasu et al [49] | Physicians | Factors associated with telemedicine practice included having at least 5 years of work experience, undergoing senior residency training, receiving formal telemedicine training, possessing good knowledge of telemedicine, and having a positive attitude toward it. | Challenges identified included knowledge and skill gaps, slow internet connectivity, unstable electricity, and inadequate equipment. |
| Obi-Jeff et al [35] | Community members (90%) | Involvement of community leaders and health workers | Lack of awareness, low literacy levels, low mobile phone ownership |
| Zayyad & Toycan [46] | Health workers | Perceived usefulness, attitude, and willingness of healthcare workers | Lack of motivation, poor organizational and management policies, low experience in using tech, low literacy level |
| Ojo et al [36] | PLHIV | Factors influencing the respondent's willingness to receive the intervention were older age (OR = 0.05, 95%CI: [0.01–0.24]), having formal education (OR = 7.12, 95%CI:[3.01–16.53]), being diagnosed over 10years ago (OR = 15.63, 95%CI:[3.02–80.83]) and previous use of phone to send text messages, access the internet (OR = 2.2, 95%CI:[1.2–3.9] | NR |
| Iliyasu et al [48] | PLHIV | Acceptance of intervention was higher among males (AOR)=1.58, 95% CI = 1.12–3.72), participants with at least secondary education (AOR=1.47, 95% CI=1.27–4.97), monthly income ≥30,000 Naira (AOR=2.16, 95% CI=1.21–7.31), currently married (AOR=3.26, 95% CI=1.16–5.65), and participants without comorbidities (AOR=2.03, 95% CI=1.18–4.24). | Barriers included distrust of technology (61.9%, n=260) and privacy concerns (37.1%, n=156). |
| Anibueze et al [12] | Dental surgery patients | Experience with use of telemedicine | NR |
| Sarfo et al [41] | Stroke-free persons | Younger age, tertiary education (AOR=2.16), secondary education (AOR=2.52), semi-skilled (AOR=2.68), knowledge of history of disease (AOR=1.44), | Lack of family disease history |

NR: Not Reported; AOR: Adjusted Odds Ratio; PLHIV: Persons Living with HIV; CMD: Chief Medical Director; CMAC: Chief Medical Advisory Committee.

**Table 3. Barriers and facilitators of telemedicine in Nigeria across institutional, human-resource, technical, and financial domains.**

| Outcomes | Domains | | | |
|---|---|---|---|---|
| Barriers | Institutional (n=7) | Human-resource (n=4) | Technical (n=8) | Financial (n=3) |
| | Administrative rigidity, lack of incentive to use telemedicine, poor organizational policies, absence in school curriculum, the use of paper records, lack of government support, lack of regulation. | knowledge and skills gap, lack of trust, lack of access to information, low literacy level | Power outages, poor internet connectivity, unfriendly user interphase, lack of software, data security, distrust of technology, lack of technical support status, low mobile device ownership | Lack of monetary motivation, patient willingness to pay, and cost. |
| Facilitators | Institutional (n=4) | Human-resource (n=7) | Technical (n=6) | Financial (n=3) |
| | Aligning with state priority, supportive policy environment, privately managed organizations, presence of performance expectancy. | Involvement of community leaders, education, formal training on telemedicine, reduced workload, experience with use of telemedicine, age, marriage | Use of low-tech educational networks, access to health-related information, accessibility to internet, owning a mobile device, ease of use of platform, a mix of live and recorded interactions | Income above 20,000 and 30,000 among end-users. |

[27] and the use of paper records [33] were the identified institutional barriers to telemedicine among healthcare providers. Eight studies provided human-resource-related barriers to telemedicine among healthcare providers. Human-resource barriers included were majorly knowledge and skills gaps [11,27,40,46,49], understaffing [33], lack of trust [42] and lack of access to information [44]. Eleven studies described the technical barriers to telemedicine. Power outages [38,40,42,44,49], poor internet connectivity [11,44,49], application not user friendly [26,33], lack of software [50] and data security [25]. The financial barrier was a lack of monetary motivation [50].

Facilitators of the use of telemedicine among healthcare providers were institutional, human-resource-related, and technical-related. Working in privately funded organizations as well as the presence of performance and effort expectancy were the identified institutional facilitators [24,26]. Human-resource-related facilitators were detailed in six studies. These facilitators included awareness of telemedicine [25,38–40,46,49], education and formal training, and reduced workload [34]. Technical facilitators were reliable in the system [40], the use of low-tech educational networks [44] and access to health-related information [42].

**Barriers and facilitators of telemedicine among policymakers in Nigeria.** Barriers to the provision of telemedicine among policymakers were institutional, human-resource, and technical related. Lack of regulation or initiative in place for telemedicine process [29] and lack of government support were the stated institutional barriers [37]. Low literacy level was the major human-resource barrier [34,35]. Three studies presented the technical barriers to telemedicine among policymakers. Identified technical barriers were the perception of low mobile phone ownership among end-users [34,35], poor internet connectivity, and inconsistent electricity [29]. Facilitators of the implementation of telemedicine among policymakers were institutional and human-resource-related. Institutional facilitators were aligning telemedicine services with state-priority [34] and supportive policy environment [28] while human-resource-related was mostly involvement of community leaders in supporting the deployment of telemedicine services [35].

### Occurrence of reported barriers and facilitators across broad domains

Among end users and patients, the most reported barriers were technical while facilitators were majorly technical and human-resource-related (Fig 2). Similarly, among policymakers, the most reported barrier was technical, and the facilitator was mostly institutional. Also, technical, and institutional elements were mostly barriers to telemedicine utilization among healthcare providers while facilitators were technical and human-resource-related.

### Quality assessment

Quality assessment was performed on cross-sectional and mixed methods studies using the JBI Criteria for Analytical Studies. Studies with three entries of 'No' or 'unclear' were classified as low quality consistent. Q6 and Q7 on identification

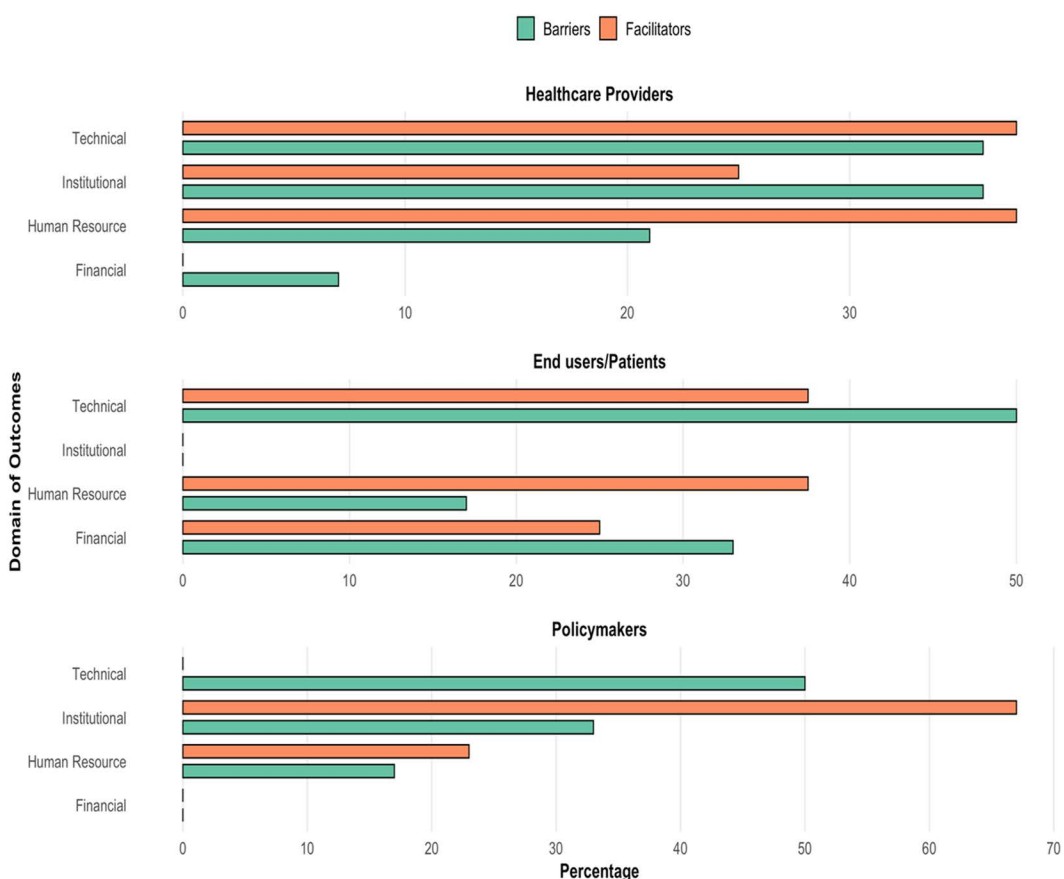

**Fig 2. Occurrence of barriers and facilitators of telemedicine in percentages across the domains of technical, institutional, human resource, and financial factors per responses of the study population in the included studies.**

and resolution of confounding respectively did not apply to the included studies and hence were not captured. All 19 cross-sectional and three mixed-method studies were considered to have a low risk of bias (Table 4).

For the seven qualitative studies, quality assessment was performed using the JBI Criteria for Qualitative Research. Most of the qualitative studies (n = 5) were low risk (S2 Appendix). The common quality issue was sample size and under-representation of the participants and in all four of the studies, the authors acknowledged this as a limitation [30,34,35,42].

## Discussion

To the best of our knowledge, this is the first systematic review to synthesize evidence on the barriers and facilitators of telemedicine in Nigeria. Most of the included studies were cross-sectional studies, conducted mostly in the southern parts of Nigeria, and published in the last five years. Findings from our study showed that generally there were more distinct barriers in the technical domain than in any of the other domains. Technical barriers reported in our review included issues with power outages or inconsistent electricity, poor internet connectivity, unfriendly or complex user interphase, and lack of software. Others were data security, distrust of technology, lack of technical support, and low mobile device ownership.

Our review also showed that the most reported technical barrier is power outages or inconsistent electricity. Limited access to electricity among health facilities in SSA countries has been identified to be a major challenge to healthcare delivery in the region. Findings show that on average over one-quarter of all health care facilities in SSA lack access to

**Table 4. Quality appraisal of cross-sectional and mixed method studies using the Joanna Briggs Institute checklist for analytical cross-sectional studies.**

| First Author | Q1 | Q2 | Q3 | Q4 | Q5 | Q6 | Q7 | Q8 | Risk of Bias |
|---|---|---|---|---|---|---|---|---|---|
| Olamoyegun et al | Y | Y | N | Y | * | * | Y | Y | L |
| Akwaowo et al | Y | Y | Y | Y | * | * | Y | Y | L |
| Noel et al | Y | Y | Y | Y | * | * | Y | Y | L |
| Archer et al | Y | Y | N | Y | * | * | Y | Y | L |
| Oyetunde et al | N | Y | N | Y | * | * | Y | Y | L |
| Onigbogi et al | Y | Y | Y | Y | * | * | Y | Y | L |
| Iliyasu et al | Y | Y | N | Y | * | * | Y | Y | L |
| Zayyad & Toycan | Y | Y | Y | Y | * | * | Y | Y | L |
| Ojo et al | Y | Y | Y | Y | * | * | Y | Y | L |
| Iliyasu et al | Y | Y | N | Y | * | * | Y | Y | L |
| Anibueze et al | Y | Y | ? | Y | * | * | Y | Y | L |
| Sarfo et al | Y | Y | Y | Y | * | * | Y | Y | L |
| Adenuga et al | Y | Y | Y | Y | * | * | Y | Y | L |
| Udenigwe et al | Y | Y | N | Y | * | * | Y | Y | L |
| Kola et al | Y | Y | ? | Y | * | * | Y | Y | L |
| Owoeye et al | Y | Y | Y | Y | * | * | Y | Y | L |
| Olunfunlayo et al | Y | Y | ? | Y | * | * | Y | Y | L |
| Nelissen et al | Y | Y | ? | Y | * | * | Y | Y | L |
| Yakubu et al | Y | Y | N | Y | * | * | Y | Y | L |
| Yakubu et al | Y | Y | Y | Y | * | * | Y | Y | L |
| Dele-Ojo et al | Y | Y | Y | Y | * | * | Y | Y | L |
| Nduka et al | Y | Y | ? | Y | * | * | Y | Y | L |

Key: Y: Yes, N: No, L: Low, H: High. Q1: Criteria for Inclusion; Q2: Description of Study Subjects and Settings; Q3: Exposure Measured in Valid and Reliable Manner; Q4: Objective or Standard Criteria for Measuring Condition; Q7: Outcome Measured in Valid and Reliable Way; Q8: Appropriate Statistical Manner; *: Not applicable;?: Not clear.

electricity, and about three-quarters lack access to reliable electricity supply [51]. In Nigeria, a power outage is a peculiar challenge. The country generates about 4500 Megawatts for a population of more than 200 million people, with an electrification rate of about 70% and more than 90% of rural communities lacking access to electricity [52]. Therefore, the provision of this critical infrastructure to support telemedicine services in Nigerian health facilities would remain a challenge as significant investment over the long term would be required. Noting that Nigeria's total health spending is 4% of her GDP, lower than the African average of 7.2%, the Lancet Nigeria Commission calls for wider and increased investment by the Nigerian government in the health sector as well as to take a nationalistic approach towards the digitisation of the health system [53].

We also observed that technical barriers were the most reported across the three broad populations of patients, healthcare providers, and policymakers. This finding is consistent with available literature that has assessed barriers and facilitators to telemedicine. For instance, our result aligns with the outcome of a systematic review that assessed the barriers to telemedicine adoption worldwide [20] It was noted that technical-specific barriers had the most occurrences among identified barriers. Similarly, another review that evaluated the barriers and facilitators of telemedicine in paediatric practice found that lack of information technology expertise and issues around internet connectivity were major barriers [54]. Generally, technical issues have been noted as a challenge to telemedicine in developing countries [55–57]. Though there are suggestions that reasonable progress has been made in SSA countries such as Burkina Faso, Ghana, and Nigeria about

integrating telemedicine into the healthcare system, there is the acknowledgment that technical-related issues remain a barrier to telemedicine adoption and implementation [18]. The WHO recognises the importance of technical factors in the implementation of telemedicine services. Thus, in its Guide to countries and technical partners on effective planning and implementation of digital interventions, the first recommended process is the assessment of the current state and enabling environment. This entails conducting an inventory of existing or previously used software applications and tech-related systems to gain insights and understand the requirements for interoperability [58].

While there are valid indications that communities and end users should be incorporated in the development of telemedicine services to capture their peculiar needs, most recommendations in the literature have indicated that education and training are vital to the resolution of most of these technical issues. Accordingly, in our review, the domain with the most occurrence of facilitators was human-resource-related with education, formal training on telemedicine, and experience with the use of telemedicine the most reported. Kruse et al. reported that globally top barriers to the adoption of telemedicine could be overcome through training on the use of telemedicine platforms. Our finding was also consistent with another review where it was reported that formal telemedicine training and educational programs could facilitate the adoption of digital technologies among healthcare professionals [59]. Similarly, from the three studies in our review that estimated the association between telemedicine utilization and education, the reported odds ratio ranged from 1.47 – 7.12, underscoring the importance of education in telemedicine use among patients and healthcare providers [41,47,49].

Among healthcare providers in Nigeria, we assume deficits in knowledge of telemedicine practice could be because of two reasons First, it could be attributed to the lack of training in medical school. El Tantawi et al. noted the lack of telemedicine in the education curriculum in Nigerian schools and this could also contribute to the poor foundation in telemedicine knowledge and practice among healthcare professionals and the consequence effect in telemedicine utilization. Second, where telemedicine training is provided, online delivery methods are most common, which typically suit busy professionals [60]. However, it is suggested that telemedicine training requires certain hands-on skills that may not be aptly captured by online education and in this regard, practical training has often provided better feedback. Thus, a systematic review capturing a global study population of 6172 students indicates that the level of knowledge of medical students with regards to telemedicine was extremely insufficient, and where there were modules on the subject topic, students tended to fail these educational courses [61]. This review aligns with existing recommendations from the literature to call for the inculcation of hands-on telemedicine training in undergraduate medical training.

Our review also provides data on the willingness to use telemedicine interventions in Nigeria. From the included studies, most persons, whether end users or providers were willing to use telemedicine, with a range of 50 – 100%, suggesting the potential for widescale adoption of telemedicine if barriers are adequately resolved. This finding is consistent with outcomes of similar studies conducted in countries where telemedicine use has been effective. For instance, in China, about 88% of surveyed physicians supported telemedicine implementation while in Iran, it was between 61–80% among specialists [62,63]. A national survey of patients' willingness to use telehealth services in the United States of America estimated the outcome to be 64.5% [64]. Some studies in parts of Africa have also shown considerable willingness to utilize telemedicine services [65,66]. Where willingness was low, ease of use was an attributable factor [67]. Thus, there is an emphasis that telemedicine should be co-designed with local communities, end users, and providers as well as tailored to the peculiar features of the users.

## Implication for telemedicine adoption

This systematic review ultimately underscores that in adopting and scaling telemedicine in Nigeria, peculiar attention needs to be given to issues such as power outages, internet connectivity, and a shortage of health professionals with technical expertise in using telemedicine platforms. Thus, the government could also lead initiatives to facilitate the development of specific power systems for tertiary health institutions and prioritise internet coverage in Nigerian health facilities through public-private partnerships. This review also identifies the lack of regulation as a major barrier to telemedicine use

in Nigeria. Notable to state that Nigeria indeed lacks an Act specifically regulating telemedicine or an agency regulating telemedicine [68]. Recognising the importance of telemedicine in increasing access to healthcare, this review first advocates for government to promulgate an act to regulate telemedicine in Nigeria. This would provide a framework to guide health facilities in the adoption of tailored telemedicine services. Also, our study identifies education and telemedicine training as key facilitators of telemedicine. Hence, we call for the training and retraining of health providers on telemedicine practice as well as the inclusion of telemedicine in the training curriculum of medical students.

Some of the included studies provided recommendations that could advance telemedicine implementation in Nigeria. These recommendations were synthesised and presented under major themes (See Table 5). Broadly, there were suggestions that the focus should be on education and sensitization of healthcare providers and patients, institutionalization of telemedicine via regulations, and cost assessments. Further, there were strong indications that involving stakeholders was critical to the successful implementation of telemedicine services.

## Implications for future research

In conducting this systematic review, we identified five areas with implications and direction for future research. First, future studies should incorporate a broader range of databases, including regional repositories and government reports. This could have the potential to find other findings which may have been overlooked in this study, further providing a comprehensive outlook on the barriers and facilitators to the provision and adoption of telemedicine in Nigeria. Second, this review noted that only 24 per cent of the eligible studies were conducted in Northern Nigeria, indicating that majority of research and practice work on telemedicine are in the southern regions of the country. Therefore, we recommend that researchers should extend more telemedicine studies to the northern part, thereby increasing representation of studies from Northern Nigeria, to gain insights into peculiar challenges, opportunities as well as barriers and facilitators of telemedicine in Nigeria. Such studies should particularly examine cultural and behavioral barriers to the end-users of telemedicine. With these, future work would address societal beliefs, trust in technology and digital literacy to develop effective telemedicine adoption strategies. Overall, this would help to capture regional variations in telemedicine adoption and provide value in targeted policies developed for telemedicine implementation across the regions of the country.

**Table 5. Table showing the recommendations of included studies for effective implementation of telemedicine in Nigeria.**

| First Author | Recommendation | Description |
|---|---|---|
| Iliyasu et al [49]<br>Onigbogi et al [39]<br>Owoeye et al [38]<br>Akwaowo et al [25] | Education of healthcare providers and patients on telemedicine | Knowledge and skills gap, lack of trust, lack of access to information, low literacy level |
| Noel et al [27],<br>Olufunlayo et al [37],<br>Adenuga et al [24],<br>Yakubu et al [11]<br>Yakubu et al [45] | Institutionalizing telemedicine | Implementation of telemedicine regulation and policy, Integration of telemedicine into the curriculum. |
| Obi-Jeff et al [34]<br>Itanyi et al [30],<br>Nelissen et al [33] | Cost assessment | Costs of implementation and care be considered in the development of telemedicine models. |
| Udenigwe et al [43],<br>Eze and Okojie [29],<br>Oyetunde et al [40],<br>Van Gurp et al [44],<br>Shekoni et al [42],<br>Yakubu et al [45],<br>Obi-Jeff et al [35] | Involvement of stakeholders, healthcare providers and end users | Enhancing interoperability of platform, involving healthcare providers in verification proves, development of web-based learning modules, addressing perception of healthcare providers, prioritizing the needs of both providers and patients as well as implementing feedback mechanism and integration of intervention with existing community structure. |

There is also a notable paucity of studies assessing feasibility costs for telemedicine adoption in Nigeria. We recommend that funders, government and policymakers consider commissioning studies to assess the economic feasibility of telemedicine services. Such studies should consider peculiarities of geographical regions and category of users of the telemedicine service provided. Appropriate and feasible funding models could also be assessed and proposed for such studies. Fourth, this review identified that Nigeria lacks an agency or act to regulate telemedicine. By implication, there is no existing guideline to tackle emerging issues in telemedicine adoption such as data security and privacy, considering the large volumes of data or 'big data' generated consequence of telemedicine use. Broadly, risk factors associated with data security could be technological, operational and environmental, with varying implications for policymakers, users and health professionals. There are advocacies for telemedicine and similar technologies be built on a foundation of patient privacy and in the reduction of cybersecurity risks as these concerns could hinder patient perception and affect telemedicine adoption [69]. Thus, government and policymakers should consider developing and proposing data security guidelines in the implementation of telemedicine services.

Lastly, in conducting this systematic review, we identified variation is study designs and reporting studies. For instance, only three of all the eligible studies reported the associations between outcome and response variables as odds ratios. These variations limited our ability to pool effect estimates and conduct a meta-analysis. Nevertheless, this study observed an increased interest in telemedicine research, evidenced by the number of publications in the last five years. Thus, there are indications meta-analysis of the studied outcomes would be feasible in the future, with improved reporting. Such a study, utilizing a meta-analysis design, would offer a clearer picture of the strength and direction of associations and help quantify heterogeneity. This would ultimately provide for generalizable conclusions with regards barriers and facilitators of telemedicine in Nigeria. Therefore, we recommend that future studies should utilize a more refined approach such as a meta-analysis to reduce heterogeneity, thereby improving the reliability of findings.

## Limitations and strengths

We acknowledge some limitations in our study. First, we aligned with an earlier assumption of the WHO that telehealth and telemedicine were synonyms and thus did not segregate recent distinct interventions such as eHealth, mHealth, and tele-education in synthesising our results. This could have the potential to exaggerate the significance of certain domains of barriers and facilitators. Second, our search utilized only three databases, and we could have missed some relevant studies from other databases, thus overlooking other key findings. Third, we had a limited number of reviewers, and this could have influenced the depth of the assessment of the bias of included studies. Fourth, we analysed all included studies regardless of sampling techniques. There were a considerable number of studies that used non-probabilistic sampling methods, and these have the potential to affect the generalizability of our results. Additionally, the inclusion of studies with qualitative and cross-sectional designs led to some heterogeneity making it difficult to compare outcomes. Lastly, we excluded gray literature to ensure reproducibility, and this would have led to the omission of relevant literature. Nevertheless, despite these limitations, the review possesses some major strengths.

A notable strength of this systematic review is that it represents the first systematic review to synthesise the barriers and facilitators of telemedicine in Nigeria. Importantly, this is the first systematic review of barriers and facilitators of telemedicine focusing on a SSA country. Further, this review generated evidence concerning facilitators and barriers to telemedicine among key populations of end users, healthcare providers, and policymakers. Key to state that though we utilized three popular databases, we did not limit our search to any period and hence we had a rich retrospective spectrum of available literature on the subject topic. Additionally, while we had a small team of reviewers for quality assessment, four reviewers committed to the screening process and three had to reach a consensus before a response was awarded for quality appraisal. Noteworthy, key terms used during the search were extracted from multiple papers and approved by all reviewers, ensuring an extensive search of the databases.

## Conclusion

Telemedicine has been widely noted as an effective intervention for increasing healthcare access and achieving SDG 3.8. In the last five years, there has been increased scholarly output about telemedicine adoption and implementation in Nigeria. There are recognisable efforts by public and private institutions to adopt telemedicine services across the country. However, barriers that are technical and institutional related hinder the successful implementation of telemedicine in Nigeria. Power outages, poor internet connectivity, shortage of health professionals with technical expertise, and absence of regulation form major barriers under the technical and institutional domains. Top facilitators to resolve these barriers are majorly human-resource-related, specifically, education and formal telemedicine training are among the most notable. With a considerable willingness among end users and healthcare providers to adopt telemedicine services, there is a huge potential for telemedicine in Nigeria. This systematic review provides a background for policy direction and potential interventions for effective telemedicine implementation in Nigeria [70,71].

## Supporting information

**S1 Appendix.** **Sample of search strategy conducted for the systematic review of barriers and facilitators of telemedicine in Nigeria across PubMed, CINAHL and Scopus databases.**
(DOCX)

**S2 Appendix.** **Quality appraisal of qualitative studies using the Joanna Briggs Institute criteria for qualitative research.**
(DOCX)

**S1 Checklist.** **PRISMA checklist.**
(PDF)

## Author contributions

**Conceptualization:** Osagie Kenneth Cole, Mustapha Muhammed Abubakar, Abdulmuminu Isah, Blessing Onyinye Ukoha-kalu.

**Data curation:** Osagie Kenneth Cole, Mustapha Muhammed Abubakar, Abdulmuminu Isah, Suleman Hayatu Sule, Blessing Onyinye Ukoha-kalu.

**Formal analysis:** Mustapha Muhammed Abubakar, Abdulmuminu Isah, Suleman Hayatu Sule, Blessing Onyinye Ukoha-kalu.

**Investigation:** Osagie Kenneth Cole, Mustapha Muhammed Abubakar, Abdulmuminu Isah, Suleman Hayatu Sule, Blessing Onyinye Ukoha-kalu.

**Methodology:** Osagie Kenneth Cole, Mustapha Muhammed Abubakar, Abdulmuminu Isah, Suleman Hayatu Sule, Blessing Onyinye Ukoha-kalu.

**Project administration:** Mustapha Muhammed Abubakar, Blessing Onyinye Ukoha-kalu.

**Resources:** Mustapha Muhammed Abubakar, Blessing Onyinye Ukoha-kalu.

**Supervision:** Blessing Onyinye Ukoha-kalu.

**Writing – original draft:** Mustapha Muhammed Abubakar, Abdulmuminu Isah, Suleman Hayatu Sule, Blessing Onyinye Ukoha-kalu.

**Writing – review & editing:** Osagie Kenneth Cole, Mustapha Muhammed Abubakar, Abdulmuminu Isah, Suleman Hayatu Sule, Blessing Onyinye Ukoha-kalu.

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
