## [Decision Letter · Decision Letter 0]

Response to Reviewers Revised Manuscript with Track Changes Manuscript
**Journal Requirements: Additional Editor Comments (if provided): Reviewers' Comments:**

**Comments to the Author**

1. Does this manuscript meet PLOS Digital Health’s publication criteria?

Reviewer #1: Yes

Reviewer #2: Yes

2. Has the statistical analysis been performed appropriately and rigorously?

Reviewer #1: I don't know

Reviewer #2: N/A

3. Have the authors made all data underlying the findings in their manuscript fully available (please refer to the Data Availability Statement at the start of the manuscript PDF file)?

Reviewer #1: Yes

Reviewer #2: Yes

4. Is the manuscript presented in an intelligible fashion and written in standard English?

Reviewer #1: Yes

Reviewer #2: Yes

Reviewer #1: 1- Future studies should incorporate a broader range of databases, including regional repositories and government reports.

Include Gray Literature and Preprints

Including government policies, unpublished studies, and conference papers could provide additional insights into ongoing telemedicine initiatives.

2-Increase Representation of Northern Nigeria

More studies should be conducted in northern and rural areas to capture regional variations in telemedicine adoption and barriers.

3-Standardize Study Selection Criteria

Using a more refined approach, such as meta-analysis or thematic synthesis, could reduce heterogeneity and improve the reliability of findings.

4-Examine Cultural and Behavioral Barriers

Future research should explore societal beliefs, trust in technology, and digital literacy to develop more effective telemedicine adoption strategies.

5-Propose Data Security Guidelines

The article should discuss potential cybersecurity risks and regulatory measures for telemedicine in Nigeria, ensuring patient confidentiality and compliance with international data protection laws.

6- Assess Telemedicine Costs and Funding Models

The study should analyze the economic feasibility of telemedicine services, including public-private partnerships and subsidy models to enhance affordability.

Reviewer #2: The paper discusses the barriers and facilitators of telemedicine in Nigeria using a systematic review of peer-reviewed papers retrieved from three databases and screened based on inclusion and exclusion criteria. The authors used PRISMA as a systemic review framework. The paper is well structured and written in standard English. The paper also presents interesting findings that could guide Nigeria’s and other developing countries’ journey towards digitization and telemedicine.

I have comments and questions that would further improve the quality of the paper.

The unavailability of page numbers makes reviewing and forwarding comments and questions page by page difficult. Please insert the page number.

Please use proper heading numbers considering subsections.

Provide references to PRISMA and Joanna Briggs Institute (JBI) Critical Appraisal Checklist.

I would restructure and label the screening section of Figure 1 based on inclusion and exclusion criteria (duplication, language, peer-reviewed, title, abstract, and full-text review)

The authors mentioned, 'this study excluded all reviews (scoping, systematic, narrative, and literature), commentaries, editorials, correspondences, and case studies.’ but failed to give their justification. Please provide a justification for why you excluded these papers.

Please discuss why interventional, experimental, and quasi-experimental studies that aimed to evaluate the feasibility of implementing telemedicine practice were excluded in this study.

The authors mentioned, ‘Conflicts were resolved by contacting a third reviewer, AI.’ Could you please discuss which AI system you have used and how you used it? Can we consider AI as a reviewer?

Figure 2 shows that patients, healthcare providers, and policymakers mentioned ‘technical, institutional, human resource, and financial factors’ as both a barrier and a facilitator with different magnitudes. Could you please discuss the issue further and clarify why and in what context it is?

I would extend the Quality Assessment section by adding the quartiles, SJR, or impact factor of the journal where the papers were published.

I would also exclude the papers that are categorized as ‘low quality’ based on the Quality Assessment from the systematic review, as those papers may not reflect the real picture of barriers and facilitators.

I would exclude the second limitation on the number of databases you searched and include other well-known databases. I wonder what restrains you from including other databases.

**Do you want your identity to be public for this peer review?** For information about this choice, including consent withdrawal, please see our Privacy Policy

Reviewer #1: No

Reviewer #2: No

**Figure resubmission:****Reproducibility:** To enhance the reproducibility of your results, we recommend that authors of applicable studies deposit laboratory protocols in protocols.io, where a protocol can be assigned its own identifier (DOI) such that it can be cited independently in the future. Additionally, PLOS ONE offers an option to publish peer-reviewed clinical study protocols. Read more information on sharing protocols at https://plos.org/protocols?utm_medium=editorial-email&utm_source=authorletters&utm_campaign=protocols

---

## [Decision Letter · Decision Letter 1]

Barriers and Facilitators of Provision of Telemedicine in Nigeria: A Systematic Review

PDIG-D-25-00109R1

Dear Dr Ukoha-kalu,

We are pleased to inform you that your manuscript 'Barriers and Facilitators of Provision of Telemedicine in Nigeria: A Systematic Review' has been provisionally accepted for publication in PLOS Digital Health.

Best regards,

Haleh Ayatollahi

Section Editor

PLOS Digital Health

**Additional Editor Comments (if provided):**

**Reviewer Comments (if any, and for reference):**

Reviewer's Responses to Questions

**Comments to the Author**

Reviewer #1: All comments have been addressed

Reviewer #2: All comments have been addressed

publication criteria?

Reviewer #1: Yes

Reviewer #2: Yes

3. Has the statistical analysis been performed appropriately and rigorously?

Reviewer #1: Yes

Reviewer #2: N/A

4. Have the authors made all data underlying the findings in their manuscript fully available (please refer to the Data Availability Statement at the start of the manuscript PDF file)?

Reviewer #1: Yes

Reviewer #2: Yes

5. Is the manuscript presented in an intelligible fashion and written in standard English?

PLOS Digital Health does not copyedit accepted manuscripts, so the language in submitted articles must be clear, correct, and unambiguous. Any typographical or grammatical errors should be corrected at revision, so please note any specific errors here.

Reviewer #1: Yes

Reviewer #2: Yes

Reviewer #1: Dear author

The article addresses an important and underexplored topic with methodological rigor. It successfully aggregates diverse barriers and facilitators to telemedicine adoption in Nigeria. However, its contribution could be strengthened by expanding its scope, deepening the analysis, and providing clearer implications for policy and implementation.

thank you

Reviewer #2: The authors thoroughly addressed all my comments and questions.

**Do you want your identity to be public for this peer review?** For information about this choice, including consent withdrawal, please see our Privacy Policy

Reviewer #1: No

Reviewer #2: No
